# A Life Dedicated to Climbing and Its Sequelae in the Fingers—A Review of the Literature

**DOI:** 10.3390/ijerph192417050

**Published:** 2022-12-19

**Authors:** Tatjana Pastor, Andreas Schweizer, Octavian Andronic, Léna G. Dietrich, Till Berk, Boyko Gueorguiev, Torsten Pastor

**Affiliations:** 1Department of Plastic and Hand Surgery, Inselspital University Hospital Bern, University of Bern, 3012 Bern, Switzerland; 2AO Research Institute Davos, 7270 Davos, Switzerland; 3Division of Hand Surgery, Department of Orthopaedics and Trauma Surgery, Balgrist University Hospital, University of Zurich, 8006 Zurich, Switzerland; 4Department of Trauma, University Hospital Zurich, 8091 Zurich, Switzerland; 5Department of Orthopaedic and Trauma Surgery, Cantonal Hospital Lucerne, 6002 Lucerne, Switzerland

**Keywords:** climbing, osteoarthritis, overuse, finger, osteophyte, load adaptation

## Abstract

Fingers of sport climbers are exposed to high mechanical loads. This work focuses on the fingers of a 52-year-old active elite climber who was the first in mankind to master 8B (V13), 8B+ (V14) and 8C (V15) graded boulders, bringing lifelong high-intensity loads to his hands. It is therefore hypothesized that he belongs to a small group of people with the highest accumulative loads to their fingers in the climbing scene. Fingers were analyzed by means of ultrasonography, X-rays and physical examination. Soft tissue and bone adaptations, as well as the onset of osteoarthritis and finger stiffness, were found, especially in digit III, the longest and therefore most loaded digit. Finally, this article aims to provide an overview of the current literature in this field. In conclusion, elite sport climbing results in soft tissue and bone adaptations in the fingers, and the literature provides evidence that these adaptations increase over one’s career. However, at later stages, radiographic and clinical signs of osteoarthritis, especially in the middle finger, seem to occur, although they may not be symptomatic.

## 1. Introduction

During climbing and its related training, one’s body weight must be held with sometimes only one finger, which places extremely high mechanical loads on the connective tissues and bones [1]. In recent years, the popularity of sport climbing has risen continuously and has received a recent boost after inclusion in the Olympics. Therefore, younger athletes are being attracted to this sport, pushing boundaries to new levels [2]. As a consequence, athletes are starting earlier, with high-intensity training adding more climbing years to their bodies [3]. Today, it is not unusual to find 16-year-old climbers in the world cup finals, which was almost impossible 30 years ago. It is therefore expected that this new generation of climbers adds a greater cumulative load to their fingers over their career. The focus of climbing-related research in the past has laid mainly on performance and acute climbing injuries. However, with the increasing popularity of this sport, concerns have arisen about the long-term effects to the human body after intensive climbing. Therefore, the aim of this work was to describe the sequalae of lifelong high-intensity climbing to the fingers in an example of an extraordinary 52-year-old elite climber and to provide an overview of the current literature in this field. It is hypothesized that, in the future, more athletes will achieve accumulative loads in their fingers that are similar to that of the participant of the current report, which might be an outlook for the future of this new generation of climbers. However, broad conclusions cannot be drawn based on a single climber.

## 2. Methods

### 2.1. Data Collection and Examination

This work was approved by the local ethics committee (Cantonal Ethics Commission Zurich, Switzerland, BASEC-Nr. 2019-00677), and the participant signed a written informed consent form.

After a detailed examination of both hands using a goniometer and a JAMAR dynamometer (G.E. Miller, Inc., Yonkers, NY, USA), the participant filled out a detailed questionnaire regarding his current climbing and training level (see Appendix A). Furthermore, questions regarding injuries, pain and stiffness during the last 6 months were asked. Questions were chosen to evaluate the current and past performance and training behavior, as well as pain and morning stiffness, which are considered clinical signs of osteoarthritis. Following that, he received standardized anterior–posterior and lateral X-ray views (Ysio wi-D system, Siemens, Erlangen, Germany) of all fingers except the thumb of both hands using a positioning devise to ensure standardized lateral X-ray images. For each phalanx of all fingers except the thumb, two measurements were obtained digitally in a lateral view (Figure 3B) according to Bollen and Wright [4]. Both inner and outer cortical widths were measured exactly in the middle of each phalanx, as previously described by Hahn et al. [5]. With these two parameters, cortical bone thickness and the medullary canal width were determined for each phalanx. Furthermore, radiological signs of osteoarthritis were rated on antero-posterior radiographs using the Kellgren–Lawrence (K–L) classification [6]. According to Kellgren and Lawrence, osteoarthritis is deemed present at grade 2 [6,7,8].

Following X-ray examination, a detailed sonography of all fingers in both hands except the thumb was performed. Ultrasound images were obtained using the Samsung ultrasound device (Medison HM70A, Samsung-Healthcare, with a 15 MHz hockey stick probe LS 6 and a linear field of view). All proximal interphalangeal joints (PIP) and distal interphalangeal joints (DIP) of fingers II-V on both hands were examined from the dorsal side. A transverse plane in the maximum flexion of the joint was used for the evaluation of cartilage thickness, whereas for the evaluation of osteophytes, a longitudinal plane during the full extension of the digit was used. Furthermore, the thickness of the articular cartilage of each phalangeal head was measured in the middle of the joint in the transverse plane by means of the software implemented in the ultrasound device.

For soft tissue evaluation (tendons, pulleys and palmar plates), every digit of both hands except the thumb were examined using the ultrasonographic probe at the proximal and middle phalanx. The thickness of the A2 and A4 pulleys were measured in the axial plane at the most distinct and stout location of the pulley. The flexor digitorum profundus (FDP) and flexor digitorum superficialis (FDS) tendons were visualized at their thickest point and were measured combined in the sagittal plane at the level directly below the A2 and A4 pulley. The palmar plate (PP) was only displayed in the sagittal plane, and thickness was measured at the PIP as well as the DIP joint between the highest part of the distal middle phalanx and the most convex part of the flexor tendons. All measurement principles for the soft tissue evaluation have been previously described [9,10]. Furthermore, a height- and weight-matched 40-year-old non-climbing male surgeon was evaluated for a direct comparison of the presented pictures. We waivered a perfect age matched control, because no significant increases in size (osteophytes, bones, cartilage, pulley, flexor tendons and palmar plates) over a 10-year follow-up period could be found in the fingers of non-climbers in previously published studies [11,12,13].

### 2.2. Literature Review

Two authors (To.P. and Ta.P.) independently conducted the literature search on PubMed in September 2022. The following keywords were used in combination: climbing, rock-climbing, bouldering, elite climbing, finger adaptation and finger osteoarthritis. The same authors separately performed the initial title/abstract screening. Afterwards, the full texts of the articles of interest were accessed.

## 3. Results

### 3.1. Participant Description

We investigated a 52-year-old right-dominant male elite sport climber, who started climbing at the age of 13 and who was the first in humankind who climbed an 8B (V13), 8B+ (V14) and 8C (V15) graded boulder. His height is 180 cm, and he weighs 75 kg, which corresponds to a BMI of 23.0 kg/m^2^. Table 1 provides a detailed overview of the anthropometrics and characteristics of this elite climber. Besides bilateral shoulder pain, he is an otherwise healthy non-smoker working as a climbing instructor as well as a climbing shoe manufacturer with no regular medication.

### 3.2. Clinical Investigation

In general, the fingers of the elite climber look rough and strong with bruises and callus formations as well as a thicker appearance compared with non-climbers (see Figure 1). In addition, typical arthritic changes are visible in both the PIP and DIP joints. Results of the clinical investigation are presented in Table 2.

### 3.3. Bone Adaptations

Signs of osteoarthritis, such as joint space narrowing, osteophytes, subchondral sclerosis and cysts, were found in both joints of all fingers of the climber. Intramedullary canals appear smaller, and the cortices appear thicker compared with the non-climber (Figure 2). Furthermore, the occurrence and size of osteophytes are greater when compared with the 10-year-old radiographs (Figure 2C). A detailed description of bone thickness, osteoarthrosis grade, osteophytes and the diameter of the intramedullary canal are presented in Table 3.

### 3.4. Soft Tissue Adaptations

During ultrasonography, A2 and A4 pulleys, flexor tendons, palmar plates and, surprisingly, the cartilage of DIP and PIP joints appeared thicker compared with the non-climber. Exemplified ultrasound pictures of the right digit II of the climber and the non-climbing control are displayed in Figure 3. Furthermore, a detailed thickness description of cartilage, flexor tendons, palmar plates and pulleys in all fingers of the climber is presented in Table 3.

## 4. Discussion

The aim of the current work was to present the long-term influence of high mechanical loads in the fingers of an elite sport climber. We chose this participant, as he was the first in humankind who climbed an 8B (V13), 8B+ (V14) and 8C (V15) graded boulder in the 1990s, and he is still actively climbing at high levels at the age of 52. Therefore, we hypothesized that he belongs to a small group of people with the highest accumulative loads in their fingers in the climbing scene. Furthermore, the findings of his fingers were by far the most severe in the literature and from our daily practice treating elite climbers with finger pain, which justifies this case report and might be an outlook for the future of the next generation of elite climbers. However, broad conclusions cannot be drawn based on a single climber. Finally, this work aimed to provide an overview of the current literature regarding bone and soft tissue adaptations in the fingers of climbers due to high loads as well as of that regarding the influence of these loads in the development of osteoarthritis.

### 4.1. Soft Tissue Adaptation in the Fingers

The investigated climber showed mechano-adaptations in his fingers in all investigated soft tissues (palmar plates, pulleys and flexor tendons). These findings are in line with other reports in the current literature. Thicker palmar plates, flexor tendons and pulleys compared to a non-climbing control group were described by Schreiber et al. [9]. Interestingly, all investigated soft tissue parameters were thicker compared with the baseline investigation in a 10-year follow-up study of the same investigated group that is still actively climbing [11], and therefore, a continuous build-up over the group’s climbing career was suggested by the authors. A correlation between this mechano-adaptation and anthropometry, level and climbing experience could, however, only be found for palmar plate thickness with body weight as well as with the highest redpoint level reached. Similar findings were also reported by Garcia et al., who reported thicker palmar plates, flexor tendons and palmar plates in the fingers of 20 adolescent sport climbers compared with non-climbing controls [14]. Additionally, Klauser et al. reported thicker pulleys in an MRI-based investigation of 52 extreme rock climbers [15]. Furthermore, thicker collateral ligaments and capsules were reported in the PIP and DIP joints of 20 sport climbers by Heuck et al. [16].

Besides a mechano-adaptation of the soft tissues, a decreased range of motion was found in almost all fingers of the investigated climber in the current report. Finger contractures were also reported in 21% of the investigated climbers of the German Junior National Team in a study reported by Schoeffl et al. [8]. Moreover, Logan et al. conducted a survey and reported contractures in 19.5% of their investigated climbers [17].

### 4.2. Bone and Cartilage Adaptation or Osteoarthritis?

The findings of thick cortices and small medullary canals in the phalanges of the investigated climber are in line with several other reports on bone adaptations in climbers and can be considered a well-known fact, and a positive correlation with climbing years has been found [4,5,7,12,15,18,19,20]. Moreover, the incidence of reported stress fractures in adolescent climbers is continuously rising [2,21,22], with the first reports dating back to the late 1990s [23]. These stress fractures are interpreted as an overload reaction as a sequela of increasing loads in the fingers of the athletes. Furthermore, bone adaptations have also been reported in adolescent climbers [14,19,20].

After the first reports appeared in the literature in the late 1980s, the question arose of whether these changes in the bone are adaptations to mechanical stress or beginning pathological osteoarthritic reactions. Hochholzer et al. as well as Heuck et al. reported adaptations in the fingerbones of climbers in X-ray and MRI investigations. However, these climbers had no pain, and these reactions were interpreted as mechano-adaptations to high loads during climbing [16,24]. Similar conclusions were drawn by several other research groups, who reported mechano-adaptations in the fingers of climbers without clear signs of osteoarthritis [9,25]. Besides bone reactions to mechanical loads, thicker cartilage in the PIP and DIP joints of all fingers were reported in an elite climbing group compared with non-climbing controls [10]. With further climbing years, the cartilage, however, decreased over a decade but was still thicker as compared with non-climbing controls. Furthermore, a clear correlation to pain could not be found [13], and the question of whether these findings were early signs of osteoarthritis or just mechano-adaptations could not be conclusively clarified. In contrast, several other studies have found clear signs of osteoarthritis in the fingers of climbers. In 1994, Bollen and Wright described subchondral cysts in almost 50% of a group of 36 climbers alongside clear signs of osteoarthritis (n = 2). However, clear signs of osteoarthritis were only found in climbers older than 40 years, and the authors questioned whether the young stars of the rock-climbing world of today become the gnarly handed middle-aged adults of tomorrow [4]. These results were confirmed by Allenspach et al. in 2011, who found a significantly higher risk for osteoarthritis in climbers as compared with non-climbing controls [18]. Another research group reported signs of osteoarthritis in up to 39% of climbers; however, some of them were surprisingly without complaints [26,27]. Two recently published long-term reports with elite climbing groups have provided the strongest evidence to support the hypothesis that, after an initial mechano-adaptation of the fingers to loads at early stages of one’s climbing career, a degeneration process occurs, resulting in early signs of osteoarthritis. Schoeffl et al. reported the progress of cortical reactions in 19 members of the German Junior National Team and 18 recreational climbers over a follow-up of 11 years with a follow-up rate of over 75%, and they found a correlation between intensive finger training as well as UIAA climbing level and osteoarthritic changes in finger joints [8,20]. Despite their young age, already 27% of elite climbers had clear signs of osteoarthritis in their fingers. Another research group reported bone thickness, osteophytes and degenerative changes over the course of 10 years in the fingers of 31 elite climber, with mean climbing years of 34 (range 25–42 years) and a follow-up rate of 100% [5,12,18]. The investigated elite climbing group had, by far, the most climbing years at the elite level in the current literature and demonstrated an increase in bone thickness and osteophyte size over the course of 10 years. Although osteophyte prevalence did not significantly increase, their size grew significantly over 10 years. Moreover, clear signs of osteoarthritis (K-L > 2) were found in 84% of the elite climbers; however, not all were directly associated with pain [12]. Another interesting finding in this follow-up study was the more severe effect of degenerative changes in DIP joints rather than in PIP joints, especially in the middle finger. Other studies have also reported the middle finger to be the most susceptible to degeneration [10,18]. This might be explained by high loads on the longest finger and on the DIP joints during the crimp position in which the PIP joints of digits II to V are hyper flexed, whereas the DIP joints of these fingers are hyper extended [28]. “Crimping” is used by 90% of all climbers [29] to maximize the contact area between their fingers and the rock [30] and to gain up to 8 cm of additional height, which is a clear advantage in reaching the next hold [31]. The participant of the current report is, by far, the climber with the highest cumulative stress in his fingers with the most severe degenerative changes and clear signs of osteoarthritis. Therefore, he might be considered an outlook for the future of younger high-performance climbers, demonstrating the most severe findings in both middle fingers.

Further research is necessary to follow the course of other elite climbers over their careers, especially at older ages. In addition, research may evaluate the correlation between body weight and osteoarthritis development, as higher athlete weights add more load to their fingers. A positive correlation between body weight and injuries has already been demonstrated in several reports [32,33,34,35,36]. Besides those, reports on the fingers of female athletes are not present in the current literature, as all reports focus on male athletes. Further effort is necessary to close this knowledge gap. Finally, future research should focus on preventive measures during the training of younger climbers. The crimp position in particular is suspected of promoting osteoarthritis, as it generates high forces in the DIP and PIP joints of the fingers [28]. A well-balanced training program should be developed for future generations of climbers.

## 5. Conclusions

Climbing induces mechano-adaptations of cortical bones, soft tissues and cartilage in the fingers of athletes. Furthermore, osteophytes occur with increasing accumulative loads most frequently in the longest and therefore most loaded fingers. The literature provides evidence that mechano-adaptations and osteophyte size increase over one’s career. However, at later stages, clear radiographic and clinical signs of osteoarthritis, especially in the middle finger, seem to occur, although they may not be symptomatic.

## Figures and Tables

**Figure 1 ijerph-19-17050-f001:**
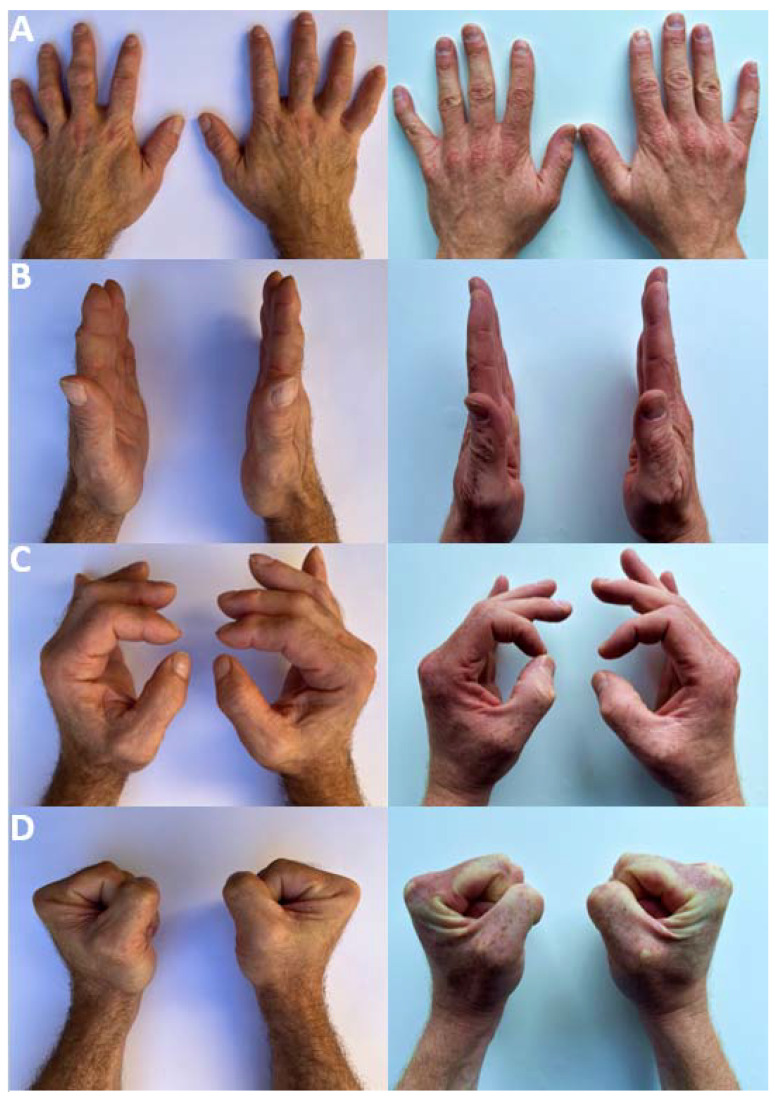
Clinical appearance of both hands of the climber (left) in contrast to hands of a non-climber (right). (**A**): View from dorsal. (**B**): View from radial with full finger extension. (**C**): View from radial with increasing finger extension for better radial view of the fingers. (**D**): View from radial with full finger flexion. Note the reduced finger flexion of the participant in contrast to the non-climber in (**D**).

**Figure 2 ijerph-19-17050-f002:**
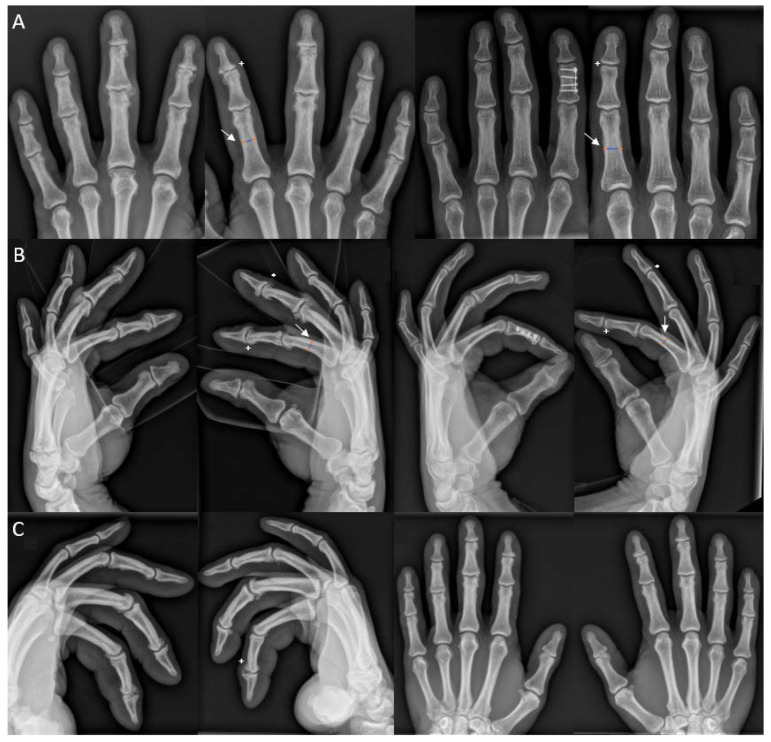
(**A**,**B**): Left: Radiographs of the participant. Right: Height- and weight-matched non-climbing male: (**A**): Anteroposterior radiographs of digits II–V of both hands. (**B**): Lateral radiographs of digits II–V of both hands. Note the thinner intramedullary canal (white arrow; blue line) and the thicker cortical bone (orange line) in the climber in contrast to the non-climbing male. Furthermore, signs of osteoarthritis (e.g., right DIP digit II marked with +) and osteophytes (*) are demonstrated in the right digit II of the climber. More severe grades of osteoarthritis and bigger osteophytes are seen in fingers II and III. A normal DIP (+) without osteophytes (*) is demonstrated in the non-climbing male with an incidental finding of a plate osteosynthesis in the middle phalanx of the left digit II after a fracture years ago. (**C**): The 10-year-old lateral radiographs of digits II-V and anteroposterior radiographs of digits II-V of both fingers of the participant. Note the evolution of osteophytes over the course of 10 years (e.g., right DIP digit II marked with (+) in (**B**)).

**Figure 3 ijerph-19-17050-f003:**
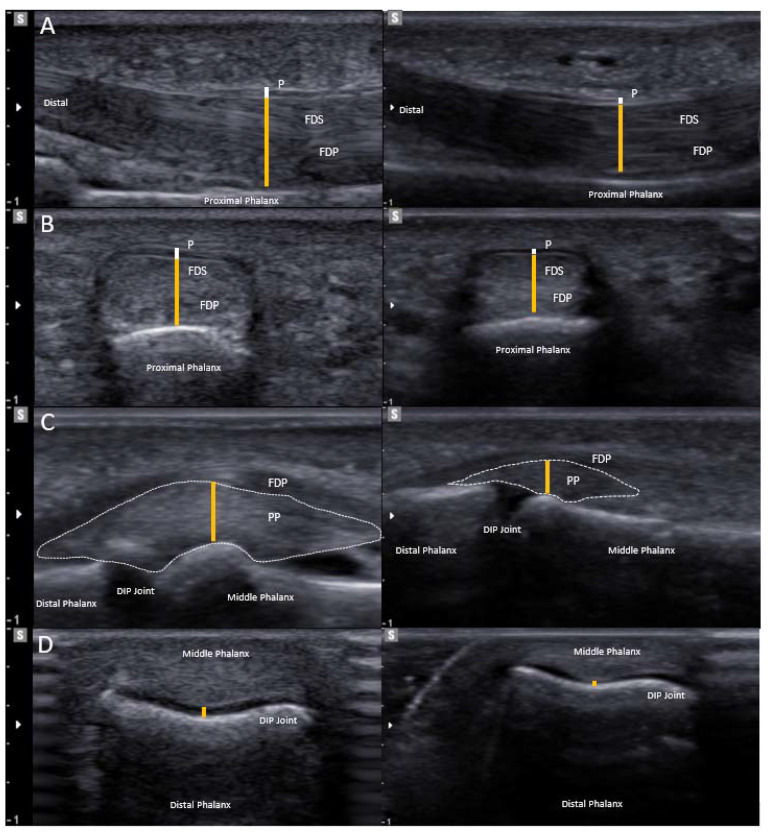
Ultrasound examination. Left: participant; Right: height- and weight-matched non-climbing male: (**A**): Longitudinal view of flexor tendons (FDS and FDP) at their stoutest point under the A2 pulley (P). (**B**): Axial view of A2 pulley (P) and the flexor tendons (FDS and FDP). Note the thicker flexor tendons (orange line) and the thicker pulley (white line) in the elite climber. (**C**): Sagittal view of the palmar plate (PP) (marked with dotted line) of a DIP joint. Note the thicker palmar plate (orange line) in the climber. (**D**): Transverse plane of the DIP joint. Note the thicker cartilage (orange line) in the climber.

**Table 1 ijerph-19-17050-t001:** Characteristics and anthropometrics of the investigated elite climber.

Variable	Value	Variable	Value
Age [y]	52	Height [mm]	180
Total years of climbing [y]	39	Weight [Kg]	75
Current climbing and bouldering [h per week]	9	BMI [Kg/m^2^]	23
Current climbing-related training [h per week]	9	Top level reached climbing (redpoint)	9a+
Training from 0–10 years of age [h per week]	0	Top level reached climbing (On-sight)	8a+
Training from 11–20 years of age [h per week]	25	Top level reached bouldering (redpoint)	8c+
Training from 21–30 years of age [h per week]	25	Current climbing level (redpoint)	9a
Training from 31–40 years of age [h per week]	20	Current boulder level (redpoint)	8b
Training from 41–50 years of age [h per week]	18	Grip strength right hand [Kg]	65
Training from 51–52 years of age [h per week]	18	Grip strength left hand [Kg]	60

**Table 2 ijerph-19-17050-t002:** Participant’s clinical investigation of digits II-V of both hands. Column 2 shows the range of motion of a non-climbing weight- and height-matched control to demonstrate normal values. Horizontal stability of the joints was assessed to test the ulnar and radial collateral ligament complex.

Structure	ROM (°) Flex/Ext Participant	ROM (°) Flex/Ext Control	Pain on Palpation	Pain during Climbing	Morning Stiffness	Horizontal Stable Joint Medial	Horizontal Stable Joint Lateral
**Right Hand**							
PIP D2	80/0/0	105/0/0	No	No	Yes	Yes	Yes
DIP D2	60/0/0	70/0/0	No	No	Yes	Yes	Yes
PIP D3	85/10/0	110/0/0	No	No	Yes	Yes	Yes
DIP D3	40/10/0	80/0/0	No	No	Yes	Yes	Yes
PIP D4	85/10/0	110/0/0	No	No	Yes	Yes	Yes
DIP D4	20/0/0	80/0/0	No	No	Yes	Yes	Yes
PIP D5	90/10/0	110/0/0	No	No	Occ.	Yes	Yes
DIP D5	40/0/0	80/0/0	No	No	Occ.	Yes	Yes
**Left Hand**							
PIP D2	85/5/0	105/0/0	No	No	Yes	Yes	Yes
DIP D2	60/0/0	70/0/0	No	No	Yes	Yes	Yes
PIP D3	85/10/0	110/0/0	No	No	Yes	Yes	Yes
DIP D3	40/10/0	80/0/0	No	No	Yes	Yes	Yes
PIP D4	85/10/0	110/0/0	No	No	Yes	Yes	Yes
DIP D4	20/0/0	80/0/0	No	No	Yes	Yes	Yes
PIP D5	90/10/0	110/0/0	No	No	Occ.	Yes	Yes
DIP D5	40/0/0	80/0/0	No	No	Occ.	Yes	Yes

ROM: range of motion according to neutral zero method; Flex: flexion; Ext: extension; PIP: proximal interphalangeal joint; DIP: distal interphalangeal joint; Occ.: occasionally; D2: Digit II; D3: Digit III; D4: Digit IV; D5: Digit V.

**Table 3 ijerph-19-17050-t003:** Descriptive values for bone thickness and medullary canal diameter at phalanx 1–3 of both hands and digits II-V, as well as cartilage thickness, osteophyte lengths, palmar plate thickness and Kellgren and Lawrence Grade of PIP and DIP of both hands and digits II-V. Furthermore, pulley and flexor tendon thickness at the level of the A2 and A4 pulley of both hands and digits II-V are demonstrated.

Structure	Cortical Thickness [mm]	Medullary Canal [mm]	Structure	Cartilage Thickness [mm]	Osteophyte Length [mm]	Kellgren and Lawrence Grade [0–3]	Palmar Plate Thickness [mm]	Structure	Pulley Thickness [mm]	Flexor Tendon Thickness [mm]
**Right Hand**			**Right Hand**					**Right Hand**		
P1 D2	6.3	2.8	PIP D2	0.5	2.0	2	3.2	A2 D2	0.7	3.5
P2 D2	5.4	2.3	DIP D2	0.1	1.2	3	2.7	A4 D2	0.6	2.3
P3 D2	3.7	1.5								
P1 D3	6.9	2.5	PIP D3	1.0	2.0	3	2.9	A2 D3	0.6	3.9
P2 D3	6.1	1.3	DIP D3	0.1	1.5	2	3.3	A4 D3	0.7	2.2
P3 D3	3.9	1.8								
P1 D4	4.8	3.0	PIP D4	0.6	2.6	1	4.0	A2 D4	0.6	3.3
P2 D4	5.2	2.0	DIP D4	0.3	0.7	2	2.1	A4 D4	0.4	2.1
P3 D4	4.0	1.2								
P1 D5	4.8	2.0	PIP D5	0.3	1.6	2	2.9	A2 D5	0.6	2.7
P2 D5	3.6	1.6	DIP D5	0.3	0.8	2	2.4	A4 D5	0.6	1.9
P3 D5	3.0	1.2								
**Left Hand**			**Left Hand**					**Left Hand**		
P1 D2	6.7	1.6	PIP D2	0.4	2.3	2	3.2	A2 D2	0.8	3.0
P2 D2	5.6	1.9	DIP D2	0.1	2.5	3	2.6	A4 D2	0.7	1.5
P3 D2	4.3	1.4								
P1 D3	7.7	2.4	PIP D3	0.7	3.4	3	2.5	A2 D3	0.6	3.9
P2 D3	6.3	2.3	DIP D3	0.2	0.9	2	3.4	A4 D3	0.7	2.3
P3 D3	3.4	2.0								
P1 D4	5.3	2.0	PIP D4	0.5	2.9	2	3.6	A2 D4	0.7	3.0
P2 D4	5.2	1.7	DIP D4	0.2	0.7	2	2.5	A4 D4	0.5	2.2
P3 D4	3.7	1.3								
P1 D5	5.4	1.4	PIP D5	0.4	1.6	2	3.3	A2 D5	0.6	2.6
P2 D5	3.9	1.5	DIP D5	0.3	1.0	1	2.4	A4 D5	0.5	1.7
P3 D5	3.3	1.7								

P1: phalanx 1; P2: phalanx 2; P3: phalanx 3; PIP: proximal interphalangeal joint; DIP: distal interphalangeal joint; A2: level of A2 pulley; A4: level of A4 pulley; D2: dig II; D3: dig III; D4: dig IV; D5: dig V.

## Data Availability

All data relevant to this work are included in the article or are uploaded as Appendix A.

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
