# Peer review of "A Life Dedicated to Climbing and Its Sequelae in the Fingers—A Review of the Literature"

_ijerph, 2022, doi:10.3390/ijerph192417050_

Round 1

Reviewer 1 Report

Thank you for this interesting case and review article. The x-ray and ultrasound images along with the descriptive methodology used to obtain measurements such as cartilage and tendon thickness, among other metrics, makes this paper particularly interesting. My single critique is that I really don't think we can infer the future finger health of young elite climbers today based on one case. The statement, "Therefore, he might be considered as an outlook to the future of younger high-performance climbers" goes a bit too far for me. 

There are many factors that can't be accounted for (is this climber genetically more predisposed to arthritic changes, has this climber utilized climbing training techniques outside the norm, do young climbers today climb more safely, are young climbers today monitored more closely, etc. etc.) I would consider explicitly adding that caveat - that we can not draw broad conclusions based on a single climber.

Otherwise I have only minor suggestions listed below. Thanks for this contribution to the climbing literature! 

line 30 "extreme" to "extremely" (adverb)

line 46 "in history" can be eliminated, is redundant

line 79 "transversal" is not a term familiar to me - I am used to the term "transverse" when describing ultrasound orientation

line 102 "cornea" is not a term familiar to me (outside of the eye) - I think this is suggesting "corns"?

line 238 "sings" should be "signs"

Author Response

please see attached point by point response 

Reviewer 2 Report

The paper of Schweizer et al. is a case-report that shown fingers adaptations occurs in a 52 years old elite climber. Overall the paper is well-written but my main concern is that I don't really see why the results of this work is of interest for an international audience. 

Below some comments:

-The participant description should be moved to the results section.

-Lines 60-63, “The participant filled out a detailed questionnaire regarding his current climbing and training level” and Furthermore, questions re-62 garding injuries and pain during the last 6 months were asked”. Did the authors use a validated questionnaire? If not, how the authors have chosen the questions and why? In addition, authors should show the questionnaire in the text or in suppl. Material. 

-Table 1: if the climber is 52 years old, how the authors could say that the training from 51-60 years of age is 18h per week? 

-How the authors have chosen the non-climber for the comparison? What is his age? Does his work include a load on his hands/fingers?

Author Response

  1. The paper of Schweizer et al. is a case-report that shown fingers adaptations occurs in a 52 years old elite climber. Overall the paper is well-written but my main concern is that I don't really see why the results of this work is of interest for an international audience. 

Reply: Thank you very much. The results might not be interesting for an international audience in a sports medicine journal. However, as climbing sport has increased in popularity within the past few years it is particularly interesting for the global climbing community. Therefore, we do believe it perfectly fits into the 2nd Edition of Mountain Sports Activities of IJERPH.

Below some comments:

  1. The participant description should be moved to the results section.

Reply: Thank you. We moved the participants description together with table 1 into the results section

Correction: Participant description moved to results

Lines 60-63, “The participant filled out a detailed questionnaire regarding his current climbing and training level” and furthermore, questions regarding injuries and pain during the last 6 months were asked”.

  1. Did the authors use a validated questionnaire?

Reply: Thank you for pointing this out. We did not use a validated questionnaire

  1. If not, how the authors have chosen the questions and why?

Reply: There is no questionnaire available in the current literature to evaluate climbers, therefore, we choose to design the used questionnaire based on interesting findings for the climber’s community and training behavior. Of course, a validated questionnaire like the Dash, Quick-Dash or SF-36 could have been used. However, these questionnaires do not achieve causal statements about each of the participant fingers but rather rate an overall condition of the complete upper extremity which do not add any value to our case as the investigated climber is still performing climbing on an elite level and is not disturbed in his everyday routine. Furthermore, the aforementioned questionnaires are susceptible to ceiling effects in “near to healthy individuals”.

Correction: LL 55 Questions were chosen to evaluate the current and past performance and training behavior as well as pain and morning stiffness, which are considered as clinical signs of osteo-arthritis

  1. In addition, authors should show the questionnaire in the text or in suppl. Material. 

Reply: Thank you. We included the used questionnaire as supplemental material. Please find the questionnaire at the end of this document

Correction: LL 52 (see supplemental material).    

  1. Table 1: if the climber is 52 years old, how the authors could say that the training from 51-60 years of age is 18h per week? 

Reply: You are absolutely right. We corrected the according section in Table 1.  

Correction: Table 1: Age 51-52

  1. How the authors have chosen the non-climber for the comparison? What is his age? Does his work include a load on his hands/fingers?

Reply: The 40-year-old “control” (which is not really a control but more an example for the reader) was chosen based on his gender, height and weight as well as his profession as manual worker (surgeon). Moreover, he represents a well-balanced average (bone thickness, cartilage thickness, tendon, pulley and palmar plate thickness) out of a group of 15 investigated non-climbing controls.

The fact that he is not aged matched may be criticized, however, in a 10year follow up study none of the non-climbing controls showed a relevant increase in all measured thicknesses (bone, cartilage, palmar plates, pulleys tendons). (Reference Nr.11, 19 & 25)

Correction: LL84: Furthermore, a height and weight matched 40-year-old non-climbing male surgeon was evaluated for a direct comparison of the presented pictures. We waivered a perfect age matched control since no significant increases in size (osteophytes, bones, cartilage, pulley, flexor tendons and palmar plates) during a 10 year follow up period could be found in the fingers of non-climbers in previously published studies (11,19,25).

Reviewer 3 Report

This is a significant contribution to the field, but I have some comments to the article. The  overview of current literature in this field should normally be presentated in the introduction and the aim of the study should have been derived from the this. The methodes seems to be relevant . The discussion should be based on the litterature in the introduction.

Figure 1could have been better explained.

Author Response

  1. This is a significant contribution to the field, but I have some comments to the article.

Reply: Thank you very much

  1. The overview of current literature in this field should normally be presentated in the introduction and the aim of the study should have been derived from the this. The methodes seems to be relevant. The discussion should be based on the litterature in the introduction.

Reply: Thank you for this point. In a basic research article we absolutely agree. However, this work is designed as a case report with a subsequent narrative review of the current literature. Therefore, we choose to keep the introduction as short as possible and discuss the relevant key points with the current literature in the discussion section.

Correction: none

  1. Figure 1could have been better explained.

Reply: Thank you. We added a better description to Figure 1

Correction:

Figure 1. Clinical appearance of both hands of the climber (left) in contrast to hands of a non-climber (right). A: View from dorsal. B: View from radial with full finger extension. C: View from radial with increasing finger extension for better radial view of the fingers. D: View from radial with full finger flexion. Note the reduced finger flexion of the participant in contrast to the non-climber in D.

Round 2

Reviewer 2 Report

The authors revised the manuscript well but my main concerns are still there.